# Factors associated with HIV viral suppression among adolescents in Kabale district, South Western Uganda

**Tugume Peterson Gordon**[1]\*, **Muhwezi Talbert**[1], **Maud Kamatenesi Mugisha**[2], **Ainamani Elvis Herbert**[1,3]

**1** Department of Public Health, Faculty of Health Sciences and Nursing, Bishop Stuart University, Mbarara, Uganda, **2** Department of Ethnobotany, Bishop Stuart University, Mbarara, Uganda, **3** Department of Mental Health, Kabale University School of Medicine, Kabale, Uganda

\* gtugume@pedaids.org

**Data Availability Statement:** All relevant data are within the manuscript and its Supporting Information files.

## Abstract

### Background

The goal of antiretroviral therapy is to achieve sustained human immune deficiency virus (HIV) viral suppression. However, research on factors associated with viral load suppression among adolescents in low and middle-income countries is limited. The objectives of this study were to determine HIV viral suppression levels among adolescents in Kabale district and the associated clinical, adherence and psychosocial factors.

### Methods

Cross-sectional and retrospective cohort study designs were used. Two hundred and forty-nine adolescents living with HIV that attended clinics between September and October 2019 at nine health facilities were interviewed and their medical records reviewed. A data abstraction tool was used to collect clinical data from adolescent's clinical charts, face to face interviews were conducted using semi-structured questionnaire adopted from the HEADS tool and in-depth interviews conducted with ten key informants. Qualitative data was analyzed using thematic content analysis. Logistic regression was used to determine the magnitude by which clinical and psychosocial factors influence viral load suppression. Odds Ratios (ORs) were used for statistical associations at 95% confidence interval considering statistical significance for p-values less than 0.05. Qualitative data collected from Key informants to support our quantitative findings was analyzed using thematic content analysis.

### Results

HIV viral suppression among (n = 249) adolescents was at 81%. Having no severe opportunistic infections was associated with viral load suppression among adolescent living with HIV (OR = 1.09; 95%CI [1.753–4.589]; p<0.001) as well as having no treatment interruptions (OR = 0.86; 95% CI [2.414–6.790]; p = 0.004). Belonging to a support group (OR = 1.01; 95% CI [1.53–4.88]; P = 0.020), having parents alive (OR = 2.04; 95% CI[1.02–4.12]; P = 0.047) and having meals in a day (OR = 5.68; C.I = 2.38–6.12, P = 0.010), were

**Funding:** The author(s) received no specific funding for this work.

**Competing interests:** The authors have declared that no competing interests exist.

**Abbreviations:** AIDS, Acquired Immune Deficiency Syndrome; ART, Antiretroviral Therapy; AYPLHIV, Adolescents and Young People Living with HIV; CD4, Cluster of Differentiation 4; HIV, Human Immunodeficiency Virus; HMIS, Health Management Information Systems; MOH, Ministry of Health; PEPFAR, Presidential Emergency Plan for AIDS Relief; SRH, Sexual Reproductive Health; TB, Tuberculosis; UNAIDS, United Nations Program on HIV and AIDS; UPHIA, Uganda Population HIV Impact Assessment; USAID RHITES SW, United States Agency for International Development Regional Health Integration in South Western Uganda; USAID, United States Agency for International Development; VLS, Viral Load Suppression; WHO, World Health Organization.

significantly associated to viral load suppression. The findings also indicated that long distances from health facilities, transport challenges and unprofessional conduct of health workers that make adolescent unwelcome at health facilities negatively affected viral suppression among adolescents.

## Conclusion

The findings indicate that HIV viral suppression among adolescents on ART was at 81%. Kabale district was likely not to achieve the third 90 of the UNAIDS 90-90-90 global target for this population category. The findings further indicate that having no severe opportunistic infection and no treatment interruptions, good nutrition status, peer support and support from significant others, were highly associated with viral load suppression.

## Background

Adolescents and young people represent a growing number of people living with HIV worldwide [1]. The mortality of adolescents living with HIV due to acquired immune deficiency syndrome (AIDS)-related causes is higher than in adults [2]. Globally, up to 150 adolescents die every day due to AIDS-related illnesses [3]. In 2016, 91% of adolescent deaths worldwide were reported in sub-Saharan Africa, and the rate of AIDS-related deaths among this age group had not reduced [1]. In addition, by 2018 children and adolescents from different parts of the world had worse viral suppression outcomes than adults [4].

The goal of ART is to achieve a sustained suppression of HIV replication. World Health Organization (WHO) recommends individual-level viral load as a measure of ART efficacy and treatment adherence. According to [5], treatment success is defined as a viral load threshold of <1000 copies/mL. Viral suppression is also one of the 10 global indicators in the 2015 WHO consolidated strategic information guidelines for HIV in the health sector [6]. Majority of previous studies on adolescents have been carried out in the United States and it is possible that these findings cannot be extrapolated to the African setting [7].

In 2014, the UNAIDS launched a global response toward achieving "the third 90" in the 90-90-90 targets, an initiative to end the AIDS epidemic as a public health threat by 2030 [6]. The UNAIDS' 90-90-90 campaign aims to have 90% of people living with HIV to know their status, have 90% of people living with HIV who know their status start or maintain their treatment, and have 90% of people on ART to be virally suppressed by 2020. High rates of viral suppression are attainable at the community and country level and studies have identified numerous individuals, interpersonal, and institutional factors that are associated with viral suppression in general population samples [8, 9]. However, factors associated with viral suppression among adolescents are not well document in lower- and middle-income countries.

Adolescents infected with HIV through high-risk behaviours have less optimal response to ART with only 24% achieving and maintaining undetectable viral loads over 3 years [10]. Although large studies of efficacy of ART in HIV-infected adults [11] and children [12] have been conducted, relatively less data has been collected describing the virologic outcomes among adolescents on ART. According to the WHO, the number of adolescents on ART continues to increase, reflecting successful treatment of perinatal infected children, infections during early adolescence, and expanding worldwide access to ART [5]. The unique behavioral characteristics of adolescents may lead to worse adherence to ART [13], which would increase

their risk of both morbidity and drug resistance. As a result, measurement of virologic outcomes in this population is very critical.

Uganda ranked among the top 20 high burden countries contributing 5% of AIDS-related deaths among adolescents in 2014. The deaths were attributed to the late diagnosis and poor access to treatment with most perinatally infected children starting treatment later in life. Like other countries, Uganda committed to achieve the 90-90-90 targets by 2020; however, the country is still short on achieving the ambitious target with the third 90 scoring lowest along the cascade. In 2017, about 73% of the people living with HIV knew their HIV status; 67% were enrolled on ART; while almost 60% had achieved viral suppression. To identify people living with HIV who are likely to fail on treatment (including adolescents) and monitor their quality of life, the Uganda Ministry of Health recommended viral load testing for all people who are receiving ART for at least 6 months or more. Children under 15 years continued to have much lower rates of viral suppression compared to adults.

Although adherence to ART is important for viral suppression, some children and adolescents are sufficiently adherent but their viral loads remain high, while others have suppressed viral loads despite poor adherence. In Uganda, information on viral suppression among adolescents aged 10–19 years is limited. This is because data for adolescent aged 10–15 years is lamped up under the age desegregation of 0–14 years, while adolescents aged of 16–19 years are considered under the age desegregation of adults. It is therefore difficult to ascertain outcomes of adolescents on ART with the current data. This study aimed at assessing viral load suppression associated factors among adolescents in Kabale District.

## Methods

### Study design, area and population

This was both a cross-sectional and retrospective cohort study with a mixed quantitative and qualitative approaches. The study was conducted in Kabale district, Southwestern Uganda. Kabale district borders with Rubanda district in the Southwest, republic of Rwanda in the south, Rukiga district in the north and Kanungu district in the west. The district has a population of 248,700 people as per district records of June 2020. The district has three health sub districts with a total of 55 functional health facilities. The study was conducted at nine (9) public and private not for profit health facilities providing comprehensive adolescent HIV care services. The study sites were; Kabale Regional Referral Hospital, Rugarama and Rushoroza Hospitals, Kamukira, Maziba and Rubaya HC IVs, Kamuganguzi HC III, Kigezi Health Care Foundation (KIHEFO) and Buhara NGO HC III. The study was carried among HIV positive adolescents that had been on ART for at least 6 months, adolescent peers and clinicians providing adolescent HIV services at the selected health care facilities.

### Inclusion and exclusion criteria

This study included HIV positive adolescents (10–19 years) that had been on ART for more than 6 months at each of the selected health facilities. Assent was sought from care takers for adolescents aged 10–17 years and informed consent for the 18–19 year olds. Both adolescent with suppressed and unsuppressed viral loads were interviewed. Additionally, key informant interviews were conducted with adolescent peers and HIV clinicians following their informed consent. This study excluded adolescents aged 10–17 years without a parent/guardian and the 18 years' adolescents that did not consent or refused for any reason to participate in the study. Adolescents that had been transferred-in from other health facilities were also excluded because their complete clinical records could not be accessed. The study also excluded

**Table 1. Proportionate sample allocation of respondents.**

| No. | Name of Facility | No. of active Adolescent in HIV care by end of March 2019 | Proportionate Sample from each site. (x) = (a/363*252) |
|---|---|---|---|
| 1 | Kabale Regional Referral Hospital | 187 | 130 |
| 2 | Kamukira HC IV | 46 | 32 |
| 3 | Rugarama Hospital | 41 | 28 |
| 4 | KIHEFO | 24 | 17 |
| 5 | Kamuganguzi HC III | 20 | 14 |
| 6 | Maziba HC IV | 12 | 8 |
| 7 | Rubaya HC IV | 12 | 8 |
| 8 | Buhara NGO Health Centre III | 11 | 8 |
| 9 | Rushoroza Hospital | 10 | 7 |
| | **Total** | **363** | **252** |

adolescent peers and HIV clinicians that did not consent and health workers that were not actively providing adolescent HIV services.

## Sample size and sampling procedure

The sample size was estimated using a standard formula by [14], with 5% marginal error and 95% confidence interval to arrive at a sample size of 239 participants. The sample size was then adjusted for missing data, and non-response; adjusted sample size was computed as 252 respondents. Probability proportionate to size sampling was used in this study because of the varying ART clinic sizes. This was aimed at ensuring that adolescents from larger ART clinics had the same probability of being included in the sample as those in smaller. See Table 1.

Because of the limited proportionate numbers especially from the small ART clinics, consecutive sampling technique was applied. All adolescents meeting the inclusion criteria were selected as they came to the clinic, until all the required number of adolescents at each health facility was achieved. Purposive sampling was used to select clinicians providing adolescent HIV services to share their views based on HIV clinical expertise while adolescent peers shared their experience of living with HIV.

## Data collection

Two research assistants were recruited and trained on how to collect and fill out the data tools, obtaining informed consent and assent, and observation of all the ethical considerations. The research assistants conducted face to face interviews, extracted information from the patients' clinical charts and also conducted the in-depth interviews with key informants. The research assistants visited the health facilities on their respective adolescent day HIV clinics and clearly explained the purpose of the study. They established rapport with the adolescents to make them comfortable and share personal sensitive information. Informed consent was obtained and interview were conducted in a private environment that ensured the adolescents' confidentiality. It took an average of 30 minutes to complete the questionnaire. Finally, all the completed questionnaires were collected and checked for completeness.

## Ethics approval and consent

The study was approved by the Institutional Ethics Committee of Bishop Stuart University. Both verbal and informed written consent was obtained from adolescents aged 18 years and above while parents/legal guardians consented for the adolescents aged 10–17 years. The

adolescents were also informed that their individual medical records would be reviewed and information extracted with outmost confidentiality.

## Measures

**Clinical factors.**   In this study, secondary data on adolescent's current viral load and retrospective clinical-related factors such as duration of ART, side effects, nutritional status and presence of comorbidities were extracted from adolescent's clinical charts. The patient's clinical charts have client's data on key parameters monitored by the ministry of health including; ART regimens, viral load results, nutrition status of adolescents, ARV side effects, adherence levels, Septrin and isoniazid prophylaxis among others. In case of missing data in the clinical charts, the research assistant sought for support to triangulate data from primary tools and registers like the ART registers, dispensing logs, appointment books, viral load result forms and laboratory CD4/viral load registers.

**Psychosocial factors.**   To assess for psychosocial factors associated with viral suppression, face-to-face interviews were conducted using semi-structured questionnaire that was adopted from the HEADS tool. The HEADS tool is a validated interview tool that is recommended by WHO and Ministry of Health for assessment, identification and management of adolescent mental health issues, take a psychosocial history as well as find local help and resources. The letters in HEADS stand for; Home, Education, Activities and peers, Drugs and Alcohol, Suicide and depression. This tool assesses for enrollment in any support group, access to housing and food, availability of parents, disclosure of HIV sero-status, substance abuse, employment, structural barriers; clinic fees, transport costs, distance from clinic, clinic waiting time, attitude of clinic staff and religious and cultural beliefs on HIV and ART. It also explores behavioral issues, assesses for depression, presence of support systems including friends or partner.

A score of 0, 1, or 2 is assigned for each psychosocial variable and these scores are summed to obtain a total score. A higher overall score on this screening tool translates into an indication of a greater need of immediate action. The HEADS tool was found to have a sensitivity of 82% and a specificity of 87% during a Rapid Mental Health Screening Tool for Pediatric Patients in the Emergency Department [15].

**Adherence factors.**   The information on adherence factors was collected through a face to face interview questionnaire with the adolescents. The information collected included; if the adolescents were taking the ARVs, person picking the ARV refills, ARV regimens, number of pills taken per day and frequency, storage of ARVs and whether the adolescent was taking the ARVs from school or home.

**Qualitative data.**   For qualitative data, in depth interviews were conducted with 10 key informants. The key informants included; six health workers that were purposively sampled among clinical officers and nursing officers that were active in provision of adolescent HIV services. Four adolescent peers living with HIV and on ART were also interviewed. The health workers were selected to provide and share their expert views and experience in providing HIV services for adolescent living with HIV while adolescent peers were interviewed to share their life experience dealing with HIV and taking lifelong ART. The interviews ended once saturation was achieved for each category. The research assistant sought consent from the key informant to allow audio recording during the discussion process. Recordings were then transferred and saved on a computer for transcription and analysis. Table 2 provides the details for the key informants.

## Data analysis

**Quantitative analysis.**   The quantitative data was cleaned, edited, coded and entered into the computer using Epi Data and then exported to STATA v14 for analysis. Continuous

**Table 2. Participants for in-depth interviews.**

| Health Facility | No. of Key Informants | Cadre | Sex | Age in years |
|---|---|---|---|---|
| Kabale Regional Referral Hospital | 4 | Senior Clinical Officer | Male | 44 |
| | | Clinical Officer | Female | 38 |
| | | Adolescent Peer | Male | 15 |
| | | Adolescent Peer | Male | 18 |
| Rugarama Hospital | 2 | Nursing Officer | Female | 52 |
| | | Adolescent Peer | Female | 17 |
| Kamukira HC IV | 2 | Enrolled Nurse | Female | 38 |
| | | Adolescent Peer | Male | 18 |
| Rubaya HC IV | 1 | Counselor | Female | 26 |
| Kamuganguzi HC III | 1 | Clinical Officer | Female | 34 |
| **Total** | **10** | | | |

variables such as age were categorized accordingly [16] and proportions and percentages were used for categorical variables. Analysis was conducted at three levels; univariate, bivariate and multivariate levels. Bivariate analysis was conducted to determine association between viral load suppression and independent variables. Odds Ratios (ORs) were used for statistical associations at 95% confidence interval considering statistical significance for p-values less than 0.05.

**Qualitative analysis.** The audio recordings from key informants were listened to carefully and transcribed, and to verify the quality of transcription, the tapes were double transcribed. The data was analyzed by grouping related responses into themes and subthemes in accordance with the study objectives. Key statements were quoted verbatim to give the exact meaning using the narrative qualitative analysis approach [17].

## Results

### Participants demographic characteristics

Results indicate that overall, majority (67.5%) of the respondents were aged 15–19. There was almost an equal proportional allocation for both male (124) and females (125). A large proportion (98.8%) were found to have been single with only 1.2% combined proportion falling within other marital status categories. Most (52.2%) respondents had attained at least primary school level of education followed by secondary at 40.2% with the least proportion being tertiary (3. 2%). Majority of the teenagers (95.2%) interviewed were unemployed. Table 3 shows the respondent background characteristic for overall and when disaggregated by gender.

### Proportion of adolescents who attained HIV viral load suppression

Overall, majority (81%) of the respondents had attained Viral Load Suppression (< 1000copies/ml) at the time of the interview. Fig 1 shows the proportion of adolescents who attained Viral Load suppression.

### Clinical factors associated with HIV viral load suppression among adolescents

Results in Table 4, indicate that majority (87.1%) of the respondents who had a viral load suppression were taking ARVs for a period of five (5) years and above whereas for patients who had been on ARVs for a period of six (6) months to one (1) year, all had a suppressed viral

**Table 3. Participants demographic characteristics segregated by gender (n = 249).**

| Socio-Demographic variables | Overall | Males | Females |
|---|---|---|---|
| | n(%) | n(%) | n(%) |
| **Age Category** | | | |
| 10–14 | 81 (32.5) | 41 (33.1) | 40 (32.0) |
| 15–19 | 168 (67.5) | 83 (66.9) | 85 (68.0) |
| **Marital status** | | | |
| Single | 246 (98.8) | 124 (100.0) | 122 (97.6) |
| Married | 2 (0.8) | 0 (0.0) | 2 (1.6) |
| Divorced/Separated | 1 (0.4) | 0 (0.0) | 1 (0.8) |
| **Level of education** | | | |
| None | 11 (4.4) | 3 (2.4) | 8 (6.4) |
| Primary | 130 (52.2) | 65 (54.0) | 63 (50.4) |
| Secondary | 100 (40.2) | 50 (41.2) | 49 (39.2) |
| Tertiary | 8 (3.2) | 3 (2.4) | 5 (4.0) |
| **Occupation** | | | |
| Unemployed | 237 (95.2) | 119 (96.0) | 118 (94.4) |
| Peasant | 11 (4.4) | 5 (4.0) | 6 (4.8) |
| Salary earner | 1 (0.4) | 0 (0.0) | 1 (1.0) |

load. Results show that having no severe opportunistic infections was associated with viral load suppression among adolescents (OR = 1.09; 95% CI [0.753–1.589]; p < 0.001). In addition, adolescents who were previously on second line ARV regimen were more likely to have viral suppression (OR = 1.83; 95%CI [1.276–4.520]; p = 0.004). More so, not having been diagnosed with malnutrition was significantly associated with viral load suppression among

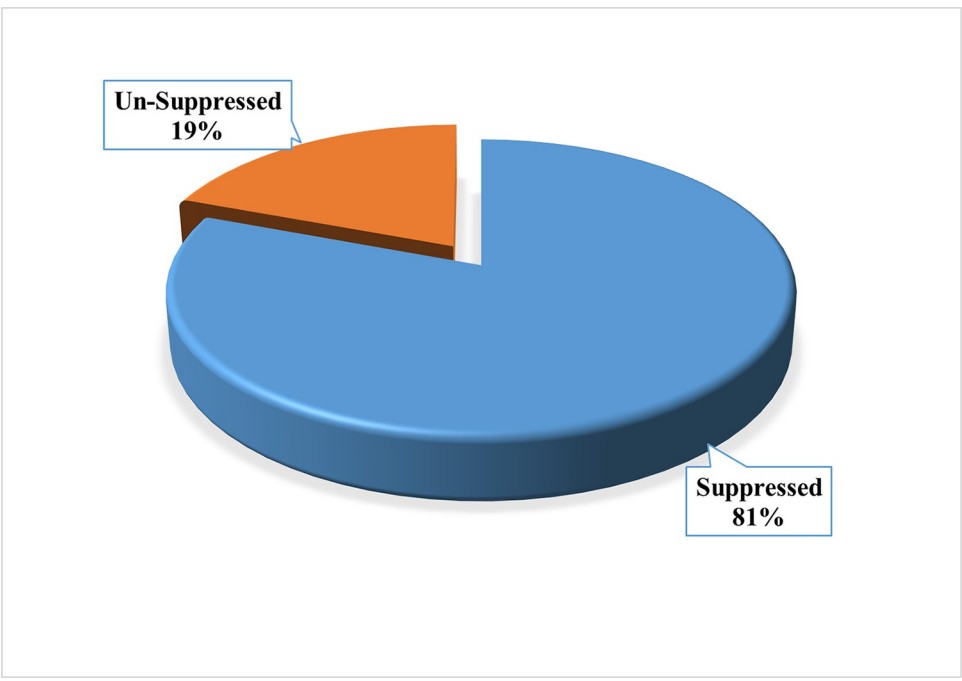

**Fig 1. Proportion of adolescents who attained viral load suppression.**

**Table 4. The association between clinical factors and viral load suppression.**

| Variable | Viral Load Suppression | | COR (95% C.I) | P-value |
|---|---|---|---|---|
| | **Yes** | **No** | | |
| | n(%) | n(%) | | |
| **Patient period on ARVs** | | | | |
| *6 months to 1yr* | 7(3.4) | 0(0.0) | 1.7 (0.99–2.92) | 0.447 |
| Greater than 1 to less than 3 yrs. | 8(4.0) | 3(6.3) | | |
| Three yrs. to less than five yrs. | 11(5.5) | 4(8.3) | | |
| Five years and above | 175(87.1) | 41(85.4) | | |
| **Severe opportunistic infection** | | | | |
| Yes | 8(4.0) | 10(20.4) | 1.09 (1.753–5.589) | **0.000** |
| No | 194(96.0) | 39(79.6) | | |
| **Current ARV regimen** | | | | |
| 1st line | 179(89.1) | 42(85.7) | 1.90 (0.574–6.286) | 0.480 |
| 2nd line | 22(10.9) | 8(14.3) | | |
| **Previous ARV regimen** | | | | |
| 1st line | 44(21.8) | 20(40.8) | 1.83 (1.276–4.520) | **0.004** |
| 2nd line | 158(78.2) | 29(59.2) | | |
| **Experience of side effects** | | | | |
| Yes | 8(4.0) | 5(10.4) | 2.01 (1.391–2.915) | 0.072 |
| No | 193(96.0) | 43(89.6) | | |
| **Ever had treatment interruption** | | | | |
| Yes | 7(3.5) | 13(26.5) | 0.86 (2.414–3.790) | **0.000** |
| No | 195(96.5) | 35(73.5) | | |
| **Diagnosed with malnutrition** | | | | |
| Yes | 0(0.0) | 3(6.1) | 0.88 (2.574–9.335) | **0.037** |
| No | 202(100.0) | 46(93.9) | | |
| **Adolescent taking Septrin/Cotrimoxazole** | | | | |
| Yes | 150(74.3) | 44(89.8) | 0.75 (2.466–8.211) | **0.007** |
| No | 52(25.7) | 4(10.2) | | |
| **Other medications including local herbs** | | | | |
| Yes | 1(0.5) | 0(0.0) | 1.48 (0.667–3.264) | 0.624 |
| No | 200(99.5) | 48(100.0) | | |

Correlation is significant at, *p <0 .05

COR is crude odds ratio

adolescents (OR = 0.88; 95%CI [2.574–9.335]; p = 0.037). Furthermore, adolescents who were taking Septrin were significantly associated with viral load suppression (OR = 0.75; 95%CI [2.466–8.211]; p = 0.007).

## Psychosocial factors associated with HIV viral load suppression among adolescents

Results on the psychosocial factors associated with viral load suppression among adolescents found out that belonging to a support group (OR = 1.01; 95%CI [1.53–4.88]; P = 0.020), having parents alive (OR = 2.04; 95%C.I [1.02–4.12]; P = 0.047]) and having meals in a day (OR = 5.68; 95%C.I [2.38–6.12]; P = 0.010), were significantly associated with viral load suppression. Other factors such as having other family members living with HIV, patient and/or caregiver disclosed about HIV status, attending school, having friends/Partner, being sexually active, use of

drugs, payment of any fee for receiving health services, distance for this facility to your home, missing attending the clinic due to lack of transport and health workers handling attitude were all insignificantly associated with viral load suppression. Details on this refer to Table 5.

## Qualitative findings

Qualitative data gathered through interviews was thematically analyzed using thematic content analysis. See Table 6.

Findings on the psychosocial factors associated with HIV viral load suppression among adolescents found out that support from significant others like family and peers, availability of care and mental health are among the psychosocial factors that determine viral suppression among adolescents.

## Theme I: Support from significant others

We found out that support from significant others like family members and peers has a significant impact on viral load suppression. This was categorized in form of family support and peer support as presented herein;

**Family support.** The study found out that a family offers accompaniment to the clinic and reminder to take medicine which is important in enhancing adherence to ART among adolescents.

*Accompaniment to clinic.* Accompaniment to clinic was found to be important in viral load suppression since it contributes much to drug adherence among adolescents on ART. This is supported by the views from one of the clinical officers who had this to say;

> *"A child who has someone to accompany him or her on clinic days, stands higher chances of adhering to drugs. This is important on viral load suppression as such a child is less likely to have problems with adherence and the possibility of skipping the days are very minimal."* (A 38-year-old female clinical officer).

*Reminder to take the medicine.* Due to pill burden, most adolescents find themselves with a challenge of skipping the medicine on some days. Getting constant reminders from family members enhances adherence leading to viral load suppression. This is in line with interview finding in which one of the participants was quoted saying;

> *"Taking medicine every other evening is a burden, at times you forget and recall in the morning. If you have someone to remind you, it becomes easier. But this is difficult at school, since we don't want other kids to learn about it."* (A 15-year-old male adolescent peer)

**Support from peers.** From the interviews, we found out that support from peers especially receiving unconditional love and belonging to a support group are important in viral load suppression.

*Showing unconditional love.* We found out that getting unconditional love from peers and friends is very important for adolescents on ART. It keeps them psychologically positive which enables them to adhere to drugs. This is supported by one of the adolescent peers who had this to say;

> *"When you have a challenge like the one we are going through and you get someone to share it with, you feel relieved. I have two friends whom always share about the future and I feel loved. I don't take anything for granted."* (An 18-year-old male adolescent peer)

**Table 5. Psychosocial factors associated with viral load suppression among adolescents.**

| Variable | Viral Load Suppression | | COR (95% C.I) | P-Value |
|---|---|---|---|---|
| | Yes | No | | |
| | n(%) | n(%) | | |
| **Belong to any support group** | | | 1 | |
| Yes | 181(90.0) | 41(85.4) | 1.01(1.53–4.88) | **0.020** |
| No | 20(10.0) | 7(14.6) | | |
| **Your parents alive** | | | 1 | |
| Yes | 90(44.8) | 20(41.7) | 2.04(1.02–4.12) | **0.047** |
| No | 111(55.2) | 28(58.3) | | |
| **Other family members living with HIV** | | | 1 | |
| Yes | 160(79.6) | 33(68.8) | 1.84(0.99–3.43) | 0.106 |
| No | 41(20.4) | 15(31.2) | | |
| **Patient and/or caregiver disclosed about HIV status** | | | 1 | |
| Yes | 198(98.5) | 48(100.0) | 0.97(0.38–6.12) | 0.615 |
| No | 3(1.5) | 0(0.0) | | |
| **Meals had in a day** | | | 1 | |
| Misses all meals in some days | 6(3.0) | 1(2.0) | 5.68(2.38–6.12) | **0.010** |
| One | 11(5.4) | 7(14.6) | | |
| Two | 129(64.2) | 31(64.6) | | |
| More than two | 55(27.4) | 9(18.8) | | |
| **Attend School** | | | 1 | |
| Yes | 180(90.0) | 45(93.8) | 0.78(0.36–8.21) | 0.280 |
| No | 21(10.0) | 3(6.2) | | |
| **Have friends/Partner** | | | 1 | |
| Yes | 39(19.4) | 13(27.0) | 1.38(1.09–6.82) | 0.240 |
| No | 162(80.6) | 35(73.0) | | |
| **Sexually active** | | | 1 | |
| Yes | 36(17.9) | 13(27.1) | 2.06(0.87–7.81) | 0.151 |
| No | 165(82.1) | 35(72.9) | | |
| **Use drugs/smokes/alcohol** | | | 1 | |
| Yes | 9(4.5) | 7(14.3) | 6.58(0.09–6.82) | 0.070 |
| No | 192(95.5) | 41(85.7) | | |
| **Pay any fee for receiving health services** | | | 1 | 0.067 |
| Yes | 0(0.0) | 2(4.1) | 8.44(0.87–8.81) | |
| No | 202(100.0) | 47(95.9) | | |
| **Distance for this facility to your home** | | | 1 | 0.859 |
| 0–14 | 108(53.7) | 26(54.2) | 0.40(0.17–4.45) | |
| 15–29 | 69(34.3) | 16(33.3) | | |
| 30–44 | 22(11.0) | 5(10.4) | | |
| 45–59 | 0(0.0) | 0(0.0) | | |
| 60+ | 2(1.0) | 1(2.1) | | |
| **Sometimes miss attending the clinic due to lack of transport** | | | | 0.067 |
| Yes | 105(52.2) | 18(37.5) | 3.37(0.87–7.81) | |
| No | 96(47.8) | 30(62.5) | 1 | |
| **Health workers treat/handle you well** | | | | 0.185 |
| Yes | 196(97.5) | 45(93.7) | 1.1 (0.6–2.2) | |
| No | 5(2.5) | 3(6.3) | 1 | |

**Table 6. Qualitative results on psychosocial factors associated with viral load suppression among adolescents.**

| Themes | Sub themes | Categories |
|---|---|---|
| Support from significant others | Family support | • Accompaniment to clinic |
| | | • Reminder to take medicine |
| | Support from peers | • Showing unconditional love |
| | | • Belonging to support group |
| Care availability | Access to health facility | • Walking long distance |
| | | • Lack of transport |
| | | • Negative treatment from health workers |
| | Nutritional care | • Lack of access to food and poor feeding |
| Mental health | Depression | • Feeling depressed |
| | | • Burden of taking daily pills |
| | Disclosure | • Fear for disclosure |

*Belonging to a support group.* It was further found out that belonging to a peer support group also has advantage towards viral load suppression. This is supported by views from one of the participants who had this to say; *"We make sure that there are adolescent peers so that those with psychosocial support issues can be linked to the peers for treatment support. We are introducing psychosocial aerial club activities so that they get involved in them whenever they come for their drug refills. This will help us to improve adherence, retention and viral load suppression." (A 44-year old Male Clinical Officer).*

## Theme II: Care availability

Availability to care was found to be another factor affecting viral load suppression among adolescents. This was categorized into access to health facility and nutritional care main factors related to viral load suppression.

**Access to health facility.** From our interviews, we found out that access to health facility has a lot impact on viral load suppression among adolescents. This was categorized into walking long distances, lack of transport and treatment from health workers.

*Walking distances.* This view is elaborated by the views from one of the participants who had this to say;

*"We have a challenge of school going adolescents who are fond of skipping appoints even to the extent of defaulting treatment. When we share with most of them we realize that the challenge of walking long distances has a hand in it. Eventually this compromises viral load suppression." (A 52-year females Nursing Officer).*

*Lack of transport.* We found out that lack of transport to health facilities contributes to challenges in care and eventually viral load suppression. This is confirmed by the views from one of the participants who had this to say;

*The distance to home to the health facility is far and you to have transport money all the time. Besides, I don't normally stay from one place, I keep moving from one place to another. This makes me skip my appointment days even when it is not intended."* (A 17-year-old female adolescent peer)

*Negative treatment from health workers.* Results further show that adolescents felt unwelcome at the health facilities because some of the health workers made adolescents

uncomfortable in getting care and treatment. This is supported by the views from one of the nursing officers who had this to say; *"Some of us health workers, behave unprofessionally. This scares away some of these adolescents and they field unwelcome at the health facility. They therefore begin the behaviors of defaulting treatment which compromises viral load suppression."* (A 38-year-old female nursing officer).

**Nutritional care.**   Nutrition is important in the life of people living with HIV/AIDS, in an event of poor nutrition, this can compromise their care and treatment. We found out that lack access to food and poor feeding is one of the issues compromising viral load suppression among adolescents.

*Lack access to food and poor feeding.* Lack of access to food and poor feeding was found out in this study among the factors compromising viral load suppression among adolescents. This is expressed by one of the participants who had this to say;

*". . . this medicine that we swallow, one of the [things] you are instructed [is] that you must eat thirty minutes before you take that drug. But then you find in that time, most times, there is nothing to eat, and so what do you do before you start, before you swallow medicine, and yet your time has approached*? *That also is a challenge we face as youth."* (An 18-year-old male peer)

## Theme III: Mental health

We found out that mental health status of adolescents has a strong influence on viral load suppression. This was categorized into depression and disclosure on the side of these adolescents.

**Depression.**   From the interviews, we found out that feelings of depression have an influence on adherence to drugs and eventually viral load suppression. This was categorized into feeling depressed and the burden of taking daily pills.

*Feeling depressed.* Results show that feeling depressed affects adolescents' adherence levels and this greatly compromises their capacity to suppress the viral load. This is supplemented by views from one of the nursing officers who had this to say;

*"By default, patients on ART should be mentally positive, anything that disorganizes them mentally, compromises adherence and viral load suppression. We have a challenge where by most adolescents present with high levels of depressive symptoms."* (A 44-year-old Male Clinical Officer)

*Burden of taking daily pills.* It was found out that taking drugs on a daily basis is a burden to most of these adolescents and they end up not swallowing them on time. This is confirmed by one of the participants who had this to say; *"You know, it is not easy packing drugs all the time, you find yourself where you have gone somewhere and time has caught you there. If there was any chance of having a drug you can swallow at least once in a week. . .."* (An 18-year-old adolescent peer).

**Disclosure.**   From the study, we found out that disclosure is another challenge affecting viral suppression among adolescents. While hiding to taking the drugs and trying to conceal their condition, they end up defaulting treatment.

*Fear for disclosure.* We found out that most adolescents have a tendency of concealing their status against their peers and significant people in their lives. This greatly affects their adherence and eventually viral load suppression. This is confirmed by the views from the participants herein;

*"If you swallow these drugs where there are people; like me, there is a lady who went on telling everyone that am positive, I don't know how she got to know but finally the people around me*

*started isolating me that I will infect them. So, the people around us make us isolate our-selves." (*An 18-year-old male peer)

## Discussion

### Proportion of adolescents on ART achieving virologic suppression

This study found that 81% of the HIV positive adolescents had achieved viral suppression. This is in line with a study conducted in South Africa which found an 81% viral load suppression among adolescents [18]. Furthermore, [19] found an 83% viral load suppression among adolescents in India. The viral suppression rate in this study is higher than that of a study done in Kenya which reported a 59% viral suppression among HIV positive adolescents [20]. In addition, a study done in South Africa found a viral suppression rate of 65.1% among adolescents [21]. Similarly, a 62.5% proportion of adolescents were virally suppressed in a study conducted in Tanzania [12]. Also, [22] revealed a 73.61% viral load suppression among HIV adolescents in Ethiopia. More so, a study by [23] found that 65.5% proportion of adolescents had achieved viral load suppression.

Similarly, [4] found a 76.8% proportion of adolescents who had achieved viral load suppression. This is in line with a study conducted in South Africa which found out that 76% of the adolescents had achieved viral suppression [7]. Systematic review on viral non-suppression rate in low or middle-income countries of the pooled estimate had 16% [24]. A lower viral load suppression of 23% was found among HIV positive adolescents in a study conducted in Uganda [25]. The overall viral suppression rate reported in this study is higher than the global viral suppression rate of 35% [20] and slightly lower than the 90% UNAIDS target. Similarly, a study by [26] found a 27.3% viral load suppression among adolescents in South Africa. In addition, 49% of the HIV positive adolescents in Brazil had achieved viral load suppression [12]. The disparities in viral suppression rates noted could be as a result of differences in systems for the management of HIV patients, including access to treatment, monitoring of adherence, and coverage, as well as disease surveillance in different regions [27]. The overall proportion of viral load suppression in the study conducted in Uganda was 73% among older adolescents [8].

The results though still below the recommended 90% by UNAIDS, there is noted progress towards achieving this target among adolescents in Kabale district. This improvement in viral suppression may be attributed better ART adherence preparation and follow by the counselors that have been deployed to support health facilities with the support of implementing partners like USAID RHITES-SW, engagement of adolescent peers in adolescent HIV services delivery, holiday comprehensive clinical and psychosocial clinics and "Ariel clubs". The health workers have also been trained in provision of adolescent friendly HIV services and separate HIV clinics for adolescent established which have reduced stigma and encouraged peer-to-peer support. The ministry of health has also provided standard service packages with standard operating procedures and guidelines developed to guide quality services delivery.

### Clinical factors associated with viral load suppression among adolescents

In contrast to the findings of this study, shorter duration on ART is associated with viral non-suppression, which is expected given the destruction of CD4 cells resulting from high viral replication. Although the relationship between viral non-suppression, immunological responses, and duration on ART is not always consistent [28]. The findings could be attributed to better ARV regimens with high treatment efficacy and tolerability, fixed drug combination with majority of the adolescent initiating on ART taking a single pill a day which improves adherence.

The findings are also in contrast to a South African study, which reported that adolescents who had been on ART between six and 12 months were more likely to have viral non-suppression (viral load > 400 RNA copies/mL) compared with those who had been on treatment longer [29]. Exposure to ART for a shorter duration was a common characteristic among adolescents who failed to achieve viral suppression. This finding is consistent with that reported in studies conducted in Northwest Ethiopia [30] and Swaziland [9].

The results of this study show that adolescents who were on second line ART were more than one times more likely to have their viral load suppressed. This study's findings agree with those of a study conducted in Zimbabwe which found that those who are on second line treatment achieved viral suppression unlike those on first line treatment [19]. Studies have demonstrated that being on second line ART is an important predictor of viral suppression among adolescents [31]. Barriers to adherence should be addressed in patients with suboptimal adherence before switching to second line therapy to improve their outcome. This study did not evaluate timeliness in switching to effective ART regimens after failure though the timeliness in switching to effective ART regimens after failure may affect viral suppression. Adolescents with delayed switching from a failing ART regimen to an effective one, experience virological failure and are at an increased risk of accumulating drug resistance and mutations [32].

Furthermore, the results indicate that odds of having suppressed viral load are almost five times higher among respondents who took their ARVs compared to those who did not take their ARVs. Similarly, [33] found that good adherence to ART was positively associated with viral suppression among adolescents. More so, adolescents with good adherence were significantly more likely to exhibit virologic suppression [34]. Good adherence to the ART is crucial for successful viral suppression, as incomplete adherence leads to an increase in HIV viremia, risk of treatment failure, and accumulating resistance mutations [35]. The findings were similar to other studies that demonstrated that suboptimal adherence to ART is associated with virological failure among adolescents [23].

Similarly, [8] demonstrated that poor adherence to ART was associated with low viral suppression. In a one to one unmatched case control study conducted in Zimbabwe, poor adherence among others was an independent risk factor for virological failure [19]. Non-adherence to medications was associated with high viral load of >1000 copies/mL in an Ethiopian study [36]. Similarly, in a study describing predictors of antiretroviral treatment adherence among a diverse cohort of adolescents' non-adherence by self-report was associated with higher viral load [37].

In addition to these studies, findings from the interviews revealed that perceived stigma, unintended disclosure and discrimination deterred the adolescents from obtaining, taking, and keeping their medications. The key informants described how the adolescents avoid going to the clinic to obtain their ART because they do not want to be seen by their peers at the clinic collecting ART medication and in so doing link them to HIV diagnosis. They further described how adolescents avoid taking pills in front of others for fear of stigma, rejection, and gossip by their peers most especially those in school. The school setting presents a unique barrier to adherence to medication for the adolescents since it not easy to leave school to keep clinic appointments for ART refill, particularly because they did not tell their teachers about their HIV status.

## Psychosocial factors associated with viral load suppression among the adolescents

In this study, it was found that a substantial proportion (81.5%) who had suppressed viral loads belonged to a support group. The findings of this study concur with those of a study

done in Tanzania which revealed that adolescents who belonged to peer support groups had better viral suppression outcomes [7]. More so, [38] also found that belonging to a support group was associated with viral suppression. This is because peer support groups have been identified as a successful method for improving the uptake of health services by adolescents thus facilitating ART adherence. Adolescents can be kept in care by using peer supporters and social workers who provide psychosocial support to improve ART adherence that results into viral suppression.

Surprisingly, a considerable proportion of the adolescents in this study reported being total orphans, though there was no statistical significance with viral load suppression. On the contrary, other studies found out that being single or double orphan was associated with virological failure [11, 39, 40]. This was similar to what some of the respondents reported in this study that the vulnerability of the adolescents as a result of the death of a biological parent or the lack of support from parents or guardians puts them at risk of having non-viral suppression.

In this study, although a significant number of the HIV positive adolescents had members in their family living with a positive HIV status, there was no significant relationship with viral suppression. However, some studies indicate that adolescents have been reported to have better viral outcomes when they had a person living with HIV in the same household [19, 41]. Furthermore, other studies conducted in Zimbabwe [42], Tanzania [28], and Ethiopia [23], found out that family support improves viral outcomes. Having HIV positive caregivers has been described as a facilitator for optimal adherence that leads to viral suppression [10]. Contrary to these studies, some health care workers reported that adolescents who live with people who are not HIV positive are likely to have poor viral outcomes.

In this study being aware of one's HIV status (disclosure) was not significantly associated with viral load suppression although most of the adolescents reported having been disclosed to. There was a very small proportion of adolescents recorded to be unaware of their own HIV status and this may have affected the findings. However, a study done in South Africa found a statistically significant association between disclosure and viral suppression among HIV positive adolescents [43]. Additionally, in Zimbabwe, disclosure increased the odds of viral suppression [19], and in Nigeria disclosure of HIV status predicted a better viral suppression as a result of good adherence [44].

Disclosed adolescents have better access to social support and tend to be less depressed over the long-term, thereby adhering to their medication well hence achieving viral suppression [42]. In addition, adolescents who have been disclosed to about their own HIV-positive status may be in a better position to access antiretroviral therapy (ART) from health facilities and psychosocial support from peer support groups which in turn translates to viral suppression [40]. However, according to a study conducted in Uganda [25], it revealed delay in disclosing HIV status to adolescents was common among the care givers and this led to non-adherence which in turn resulted into virologic failure.

Furthermore, this study showed that a significant percentage (82.4%) of the respondents that had suppressed their viral loads (< 1000copies/ml) reported not to have used drugs/ smoked/ drunk alcohol. Similarly, [19] found no difference in virological suppression and use of drugs/smoke/alcohol among HIV positive adolescents. However, a study done in low middle-income countries identified that alcohol consumption and recreational drug use contributes to virological non-suppression among adolescents as a result of poor ART adherence [24]. This was also mentioned by one of the peer adolescents that some of the adolescents who indulge in alcohol intake, have their viral loads unsuppressed.

More so, it should be noted that the independent association of alcohol consumption and virological failure is not surprising since alcohol use among adolescents has been associated with lower treatment adherence, disease progression and failed viral suppression. One study

by [45] examined the association of alcohol with viral suppression and found that those who consumed alcohol were more likely to develop virological failure compared to those who did not. In another study by [19], alcohol use was associated with poor adherence which directly translates into virological failure. The contradiction between this study and other studies could be explained by the levels of drinking alcohol among adolescents in the study, which had no effect on their drug adherence levels.

## Study limitations

This study has a number of limitations, which should be considered while interpreting the above findings. First, due to the sampling strategy, the study only included adolescents who were currently on treatment. Adolescents who were lost to follow-up were not included, which could have resulted in overestimating the rate of viral suppression. Longitudinal follow up with multiple assessments of viral load trends may provide a better picture of virological response compared to the single latest viral load measurement used in this study. Second, the study relied on self-reported data, which may be affected by recall and social desirability bias. Third, adolescents who accessed care from low volume sites providing care to less than 10 adolescents were excluded, which may bias the results given that volume of patients is a key determinant of provider competency and patient outcomes. Finally, due to a lack of genotype data for the sample, exploration of virological failure was not possible.

## Conclusions

This study finding indicate that HIV viral suppression among HIV positive adolescents on ART in Kabale district has significantly improved to 81%. However, this is still below the UNAIDS target of 90% of people on ART to be virally suppressed. The findings also indicated that severe opportunistic infection, and treatment interruptions were the major clinical factors associated with viral load non-suppression. The study also found out that support from significant others, having access to care and mental health status are highly associated with viral load suppression among the adolescents. Therefore, programs providing care to adolescents should focus on providing routine and intensive adherence counselling to prevent treatment interruptions and occurrence severe opportunistic, support disclosure and formation of adolescent peer support groups to improve viral load suppression among adolescents living with HIV.

## Recommendations

The study recommends that ministries of health and education should prioritize formal recruitment of counselors into their formal staffing structures. This will ensure sustained provision of psychosocial support to adolescent living with HIV. The study also recommends integration of HIV care services and interventions into school programs through capacity building of matrons, senior women and school nurses to provide appropriate care and support services for school going HIV positive adolescents.

There should be streamlined implementation adolescent friendly services; capacity building of health workers, opening HIV clinics over the weekends and provision of multi-month ARV refills that are re-aligned with holidays to address the needs of in-school adolescents, and integrating adolescent peers in services provision. Intensive adherence counseling should be prioritized for adolescents with non-suppressed viral loads.

Further studies also need to be carried out in the whole country with a focus on lower facilities with limited technical capacity to further answer the discrepancies in viral load suppression across health facilities and districts.

## Supporting information

**S1 Dataset.**
(XLSX)

## Acknowledgments

The authors thank the adolescents, and healthcare facility staff who participated in the study. We also thank the District Health Officer that granted permission to conduct the research at the health facilities. Special thanks go to the research assistants that were pivot in interviewing and collecting the data.

## Author Contributions

**Conceptualization:** Tugume Peterson Gordon.

**Data curation:** Muhwezi Talbert.

**Formal analysis:** Tugume Peterson Gordon.

**Investigation:** Tugume Peterson Gordon, Muhwezi Talbert.

**Methodology:** Tugume Peterson Gordon, Muhwezi Talbert.

**Resources:** Tugume Peterson Gordon.

**Software:** Tugume Peterson Gordon.

**Supervision:** Maud Kamatenesi Mugisha, Ainamani Elvis Herbert.

**Writing – original draft:** Tugume Peterson Gordon, Muhwezi Talbert.

**Writing – review & editing:** Tugume Peterson Gordon, Muhwezi Talbert, Maud Kamatenesi Mugisha, Ainamani Elvis Herbert.

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
