## [Decision Letter · Decision Letter 0]

22 Oct 2021

PONE-D-21-24368FACTORS ASSOCIATED WITH HIV VIRAL SUPPRESSION AMONG ADOLESCENTS IN SOUTHWESTERN UGANDAPLOS ONE

Dear Dr. Gordon,

Thank you for submitting your manuscript to PLOS ONE. After careful consideration, we feel that it has merit but does not fully meet PLOS ONE’s publication criteria as it currently stands. Therefore, we invite you to submit a revised version of the manuscript that addresses the points raised during the review process.

Please carefully address the reviewer comments and edit the manuscript accordingly. One of the reviewer raised substantial concerns on the study design and sampling methods. Please ensure that your decision is justified on PLOS ONE’s publication criteria and not, for example, on novelty or perceived impact.

We look forward to receiving your revised manuscript.

Kind regards,

Siddappa N. Byrareddy, PhD

Academic Editor

PLOS ONE

3. PLOS requires an ORCID iD for the corresponding author in Editorial Manager on papers submitted after December 6th, 2016. Please ensure that you have an ORCID iD and that it is validated in Editorial Manager. To do this, go to ‘Update my Information’ (in the upper left-hand corner of the main menu), and click on the Fetch/Validate link next to the ORCID field. This will take you to the ORCID site and allow you to create a new iD or authenticate a pre-existing iD in Editorial Manager. Please see the following video for instructions on linking an ORCID iD to your Editorial Manager account: https://www.youtube.com/watch?v=_xcclfuvtxQ".

Additional Editor Comments:

One of the reviewer raised substantial concerns, and therefore, I recommend authors to address carefully during the revision.

Reviewers' comments:

Reviewer's Responses to Questions

**Comments to the Author**

1. Is the manuscript technically sound, and do the data support the conclusions?

Reviewer #1: Yes

Reviewer #2: No

2. Has the statistical analysis been performed appropriately and rigorously? 

Reviewer #1: Yes

Reviewer #2: No

3. Have the authors made all data underlying the findings in their manuscript fully available?

Reviewer #1: Yes

Reviewer #2: Yes

4. Is the manuscript presented in an intelligible fashion and written in standard English?

Reviewer #1: Yes

Reviewer #2: Yes

5. Review Comments to the Author

Reviewer #1: Authors have performed a well organized retrospective study on factors associated with HIV viral suppression among adolescents. It is highly commendable that they have considered both clinical and psycho social factors into identify the factors associated with failure of achieving the goal of UNAIDS . Authors have also concluded the manuscript with favorable recommendations towards achieving the goal.

I have some minor corrections

1. an abbreviation should be used after giving the expanded version of the same For eg HIV is abbreviated in the first line of background and then expanded in second line.

2. Authors should review for grammatical errors

Reviewer #2: The manuscript by Gordoni et al investigated the HIV viral suppression levels among adolescents in Kabale District, Uganda and determined that 81% of subjects had viral load suppression. Furthermore, the authors investigated various clinical factors in the same subjects and found that the length of ART, status of opportunistic infections, and history of second line ARV regimens or antibiotics are significantly associated with viral suppression. In addition, they found that social and nutrition support are associated with viral load suppression in adolescents. Overall, the manuscript provides viral load data on a selected group of HIV-1-infected adolescents and attempts to decipher clinical and social factors impacting the viral load suppression. However, the study is poorly designed, lacks a central hypothesis, and there are a number of concerns that need to be addressed.

Major points:

1. The study lacks a central hypothesis. If the authors set out to sample the rate of viral load suppression in adolescent in the region, then the subject selection will need to be justified to represent the general HIV-1 infected adolescent population. The current subjects are skewed toward the 15-19 age group, which is unlikely to represent the general HIV-1 infected adolescent population in the region.

2. The introduction section does not have key review of the field. At least ten similar studies have already been published and cited in the discussion section, which are related to viral suppression rates in HIV-1 infected adolescents. Furthermore, the identified clinical and social factors associated with viral load suppression are largely expected. Therefore, it is unclear how the current manuscript will provide new information of interest to the scientific community.

3. Part of written informed consent are not from parent/guardian but from care providers.

4. It is unclear how the key Odds Ratios statistical association were performed for the clinical and social factors. In Tables 4 and 5, percentages of many variables look close enough between viral suppression and unsuppression groups, yet many p values have reached the significant mark. With the modest sample size, it will be more meaningful to present data in dot pot association graphs rather than the current tables.

5. The authors is over-interpreting the association data. Psychosocial (or even general psychological factors) attributes have a correlated response to lower viral suppression because of either mismanagement of treatment or default of the regiment all together. However, the authors phrase it, making it sounds like a direct influence (as in a biological factor). 

6. There are many errors in English usage and the manuscript needs to be carefully proof read.

Abstract section:  Mission preposition.  "...consecutive sampling techniques was used to select participants..."

Abstract section:  under Conclusion, they capitalized Viral when it doesn't need to be.

Background:  I'm not sure what they mean by "die owing."  Would be better to say, "die due to."  For a more technical sentence it would be more appropriate to say Adolescent's mortality is lower instead of the current vernacular.

At the end of the Background section: age disaggregation groups is an odd choice in adjective conjugation.  More appropriate to say, "This is because age groups are disaggregated into adolescents between the ages of 10-15 years..."

Last line of Sample Size and Sampling Procedures: HIV lived experiences just does not sound correct.  More appropriate to say, "...while adolescent peers shared their experience living with HIV."

Under psychosocial factors, qualitative data: They used "lived experiences" again.  It's just an odd way of saying current life experiences dealing with HIV. 

Negative treatment from health workers section: This is clearly supposed to say, "...and they feel unwelcome at the health facility."

6. PLOS authors have the option to publish the peer review history of their article (what does this mean?). If published, this will include your full peer review and any attached files.

Reviewer #1: No

Reviewer #2: No

---

## [Author Response · Author response to Decision Letter 0]

15 Dec 2021

Siddappa N. Byrareddy, PhD

Academic Editor

PLOS ONE

Dear Byrareddy

We are writing to express our appreciation to you and the team of reviewers for carefully reviewing our manuscript titled “Factors associated with HIV viral suppression among adolescents in Kabale district, southwestern Uganda’’ (pone-d-21-24368) and to resubmit it for consideration of publication in PLOS ONE. The reviewers identified aspects of the manuscript that required revision. We are pleased to resubmit to you a revised version of the manuscript, which incorporates changes that have strengthened the overall manuscript. In the attached rebuttal memo, we describe how we addressed each of the points that the reviewers raised.

Academic Editor; Please include your full ethics statement in the ‘Methods’ section of your manuscript file. In your statement, please include the full name of the IRB or ethics committee who approved or waived your study, as well as whether or not you obtained informed written or verbal consent. If consent was waived for your study, please include this information in your statement as well.

Response: We thank the editor for identifying this gap. This has been addressed in the methods section. The study was approved by the Institutional Ethics Committee of Bishop Stuart University. Both verbal and informed written consent was obtained from adolescents aged 18 years and above while parents/legal guardians consented for the adolescents aged 10-17 years. The adolescents were also informed that their individual medical records would be reviewed and information extracted with outmost confidentiality.

Reviewer #1; An abbreviation should be used after giving the expanded version of the same For e.g. HIV is abbreviated in the first line of background and then expanded in second line.

Response: We thank the reviewer for identifying this error. We have reviewed and addressed the abbreviations.

Reviewer #1; Authors should review for grammatical errors.

Reviewer #2; There are many errors in English usage and the manuscript needs to be carefully proof read. 

Response: We thank the reviewers for pointing out these grammatical errors. We have addressed the grammatical errors.

Reviewer #2; The study lacks a central hypothesis. If the authors set out to sample the rate of viral load suppression in adolescent in the region, then the subject selection will need to be justified to represent the general HIV-1 infected adolescent population. The current subjects are skewed toward the 15-19 age group, which is unlikely to represent the general HIV-1 infected adolescent population in the region.

Response: The study was conducted in Kabale district which is in south western Uganda. The word ‘’Kabale District’’ was missed in the title of the Manuscript. The study therefore did not assess viral load suppression in the whole south west region. The study subjects were selected from all the nine (9) sites providing adolescent HIV services in Kabale district and therefore the findings are representative of the district. At the time of the study a total of 363 adolescents were receiving HIV care in Kabale district. Sample size was estimated using a standard formula by (Kish & Leslie, 1965), with 5% marginal error and 95% confidence interval to arrive at a sample size of 239 participants which represented 65% of the total adolescents in HIV care. The sample size was then adjusted for missing data, and non-response; adjusted sample size was computed as 252 respondents. Probability proportionate to size sampling was used in this study because of the varying ART clinic sizes. This was aimed at ensuring that adolescents from larger ART clinics had the same probability of being included in the sample as those in smaller. The data was therefore not skewed towards the 15 -19 age group. 

Reviewer #2; The introduction section does not have key review of the field. At least ten similar studies have already been published and cited in the discussion section, which are related to viral suppression rates in HIV-1 infected adolescents. 

Response: We thank the reviewer for identifying this omission. We have revised this section to include the key review under the introduction section.

Reviewer #2; Furthermore, the identified clinical and social factors associated with viral load suppression are largely expected. Therefore, it is unclear how the current manuscript will provide new information of interest to the scientific community. 

Response: We thank the reviewer for this important question. Although we acknowledge that the findings were largely expected, not many studies have been done to find out the HIV viral associated factors among adolescent especially in the rural settings and specifically in Kabale district. These findings will also provide age specific HIV viral load suppression rates which the routinely collected HIV program data does not provide and therefore will inform designing age specific interventions. The findings will also inform the Kabale district health department on their progress on the attainment of the 90:90:90 UNAIDS target. 

Reviewer #2; Part of written informed consent are not from parent/guardian but from care providers.

Response: We thank the reviewer for identifying this. This was a wrong choice of word ‘’caregivers’’ informed consent was actually sought from the parents and guardians. We have made the corrections to use parents/guardians. e.g. ‘’ ……. parents/legal guardians consented for the adolescents aged 10-17 years’’.

Reviewer #2; It is unclear how the key Odds Ratios statistical association were performed for the clinical and social factors. In Tables 4 and 5, percentages of many variables look close enough between viral suppression and suppression groups, yet many p values have reached the significant mark. With the modest sample size, it will be more meaningful to present data in dot pot association graphs rather than the current tables.

Response: Thank you for pointing this out. It would have been interesting to explore this aspect of analysis. However, in the case of our study, we believe that it would be more appropriate to use logistic regression to generate the odds ratios at bivariate to see the relationship that exists between two variables, and multivariate to see if any, are correlated with the outcome variable than the dot plot. Furthermore, multivariate logistic regression model was used to estimate factors associated with viral load suppression adjusting for potential confounders. The multivariate logistic regression model employed a backward step wise analysis to include all the variables in the model with p< 0.05 in selecting the final model. Considering our sample size, the authors think the dot plot will become overcrowded, making it difficult to read since the points may become cluttered. In addition, a dot plot is best when the sample size is less than approximately 50. The goal in this case was to determine which variables influence or cause the outcome.

Reviewer #2; The authors is over-interpreting the association data. Psychosocial (or even general psychological factors) attributes have a correlated response to lower viral suppression because of either mismanagement of treatment or default of the regiment all together. However, the authors phrase it, making it sounds like a direct influence (as in a biological factor). 

Response: We thank the reviewer for this excellent suggestion. The authors think the psychosocial mechanisms may influence retention and adherence to treatment since issues like stress, traumatic events, and depression can lead to decreased CD4 counts and increased viral load, and therefore, accelerate HIV disease progression. In addition to the biological mechanisms that link HIV and psychological factors, social factors contribute to poor mental health outcomes among adolescents living with HIV. Because, the psychosocial problems can adversely affect adherence to ART and may lead to poor HIV treatment outcomes, the country opted for routine psychosocial factors screening as part of adherence support.

Reviewer #2; Abstract section: under Conclusion, they capitalized Viral when it doesn't need to be.

Response: We thank the reviewer for pointing out this error. We have edited the sentence ‘’ …. findings indicate that HIV viral suppression among HIV positive adolescents ……’’

Reviewer #2 Background: I'm not sure what they mean by "die owing." Would be better to say, "die due to." For a more technical sentence it would be more appropriate to say Adolescent's mortality is lower instead of the current vernacular.

Response: We thank the reviewer for suggesting a more technical sentence. There is no doubt that use of technical terms will strengthen our paper. We have rephrased the sentences to read ‘’ The mortality of adolescents living with HIV due to acquired immune deficiency syndrome (AIDS)-related causes is higher than in adults (UNICEF, 2018)’’.

Reviewer #2 At the end of the Background section: age disaggregation groups is an odd choice in adjective conjugation. More appropriate to say, "This is because age groups are disaggregated into adolescents between the ages of 10-15 years..."

Response: We thank the reviewer for providing this guidance. We have edited the sentence to read;’’ This is because data for adolescent aged 10–15 years is lumped up under the age desegregation of 0–14 years’’

Reviewer #2 Last line of Sample Size and Sampling Procedures: HIV lived experiences just does not sound correct. More appropriate to say, "...while adolescent peers shared their experience living with HIV."

Response: We thank the reviewer for this correction. We have edited the sentence to read; …….’’ while adolescent peers shared their experience of living with HIV’’."

Reviewer #2 Under psychosocial factors, qualitative data: They used "lived experiences" again. It's just an odd way of saying current life experiences dealing with HIV. 

Response: We again thank the reviewer for this correction. We have made the correction … ‘’while adolescent peers were interviewed to share their life experience dealing with HIV and taking lifelong ART’’.

Reviewer #2 Negative treatment from health workers section: This is clearly supposed to say, "...and they feel unwelcome at the health facility."

Response: We thank the reviewer for pointing out this concern. We have revised the text for the sentence to read as follows; ….’’ Results further show that adolescents felt unwelcome at the health facilities because some of the health workers made adolescents uncomfortable in getting care and treatment. 

Thank you for your time. We believe these revisions have resulted in a significantly improved manuscript.

We look forward to hearing from you in the due time regarding our submission and to respond to any further questions and comments you may have.

Sincerely,

Tugume Gordon Peterson.

---

## [Decision Letter · Decision Letter 1]

10 May 2022

PONE-D-21-24368R1FACTORS ASSOCIATED WITH HIV VIRAL SUPPRESSION AMONG ADOLESCENTS IN KABALE DISTRICT, SOUTHWESTERN UGANDA.PLOS ONE

Dear Dr. Gordon, 

Thank you for submitting your manuscript to PLOS ONE. After careful consideration, we feel that it has merit but does not fully meet PLOS ONE’s publication criteria as it currently stands. Therefore, we invite you to submit a revised version of the manuscript that addresses the points raised during the review process.

The reviewer 2, provided few minor suggestions, and therefore, I invite you to address those minor comments. 

We look forward to receiving your revised manuscript.

Kind regards,

Siddappa N. Byrareddy, PhD

Academic Editor

PLOS ONE

Journal Requirements:

Reviewers' comments:

Reviewer's Responses to Questions

**Comments to the Author**

1. If the authors have adequately addressed your comments raised in a previous round of review and you feel that this manuscript is now acceptable for publication, you may indicate that here to bypass the “Comments to the Author” section, enter your conflict of interest statement in the “Confidential to Editor” section, and submit your "Accept" recommendation.

Reviewer #2: (No Response)

2. Is the manuscript technically sound, and do the data support the conclusions?

Reviewer #2: Partly

3. Has the statistical analysis been performed appropriately and rigorously? 

Reviewer #2: Yes

4. Have the authors made all data underlying the findings in their manuscript fully available?

Reviewer #2: Yes

5. Is the manuscript presented in an intelligible fashion and written in standard English?

Reviewer #2: Yes

6. Review Comments to the Author

Reviewer #2: The authors have addressed a majority of the concerns.

Second paragraph of Background section: “Two recent studies from southern Africa reported that HIV positive adolescents had worse outcomes in terms of virological suppression and rates of virological failure.” Worse as compared to? Also the two citations?

Third paragraph of Result section. Data in this paragraph are from Table 4, but there is no reference of Table 4 in the text.

Fourth paragraph of Result section: “associated to” should be “associated with”.

Qualitative findings section: “on the on”, please remove the second “on”.

Theme I: Support from significant others section: It is unclear why the authors discussed significant others when 98.8% of the subjects were single. Furthermore, there is no specific data from Tables 5 and 6 showing that having significant others increases viral load suppression.

7. PLOS authors have the option to publish the peer review history of their article (what does this mean?). If published, this will include your full peer review and any attached files.

Reviewer #2: No

---

## [Author Response · Author response to Decision Letter 1]

17 May 2022

Reviewer #2; Second paragraph of Background section: “Two recent studies from southern Africa reported that HIV positive adolescents had worse outcomes in terms of virological suppression and rates of virological failure.” Worse as compared to? Also the two citations?

Response: We thank the reviewer for pointing out this omission. We have removed this statement since there is no comparison that was stated. 

Reviewer #2; Third paragraph of Result section. Data in this paragraph are from Table 4, but there is no reference of Table 4 in the text. 

Response: We thank the reviewer for identifying this omission. We have referenced table 4 in the text.

Reviewer #2; Fourth paragraph of Result section: “associated to” should be “associated with”.

Response: We thank the reviewer for this correction. We have edited the sentence to read; ……. “associated with’’…

Reviewer #2; Qualitative findings section: “on the on”, please remove the second “on”.

Response: We again thank the reviewer for this correction. We have made the correction and removed the second ‘’on’’ for the sentence to read ‘’…. findings on the psychosocial factors associated with….

Reviewer #2; Theme I: Support from significant others section: It is unclear why the authors discussed significant others when 98.8% of the subjects were single. Furthermore, there is no specific data from Tables 5 and 6 showing that having significant others increases viral load suppression. 

Response: We thank the reviewer for raising this critical issue. We appreciate that most significant others are persons who are important to one's well-being especially spouses or one in a similar relationship. However, in this study we also included guardian, family members and others persons including peers who provided support to adolescent including accompanying them to the clinics and provided reminders to take medicine which was important in enhancing adherence to ART.

---

## [Editor Report · Decision Letter 2]

20 Jun 2022

FACTORS ASSOCIATED WITH HIV VIRAL SUPPRESSION AMONG ADOLESCENTS IN KABALE DISTRICT, SOUTHWESTERN UGANDA.

PONE-D-21-24368R2

Dear Dr. Gordon,

We’re pleased to inform you that your manuscript has been judged scientifically suitable for publication and will be formally accepted for publication once it meets all outstanding technical requirements.

Kind regards,

Siddappa N. Byrareddy, PhD

Academic Editor

PLOS ONE
---

## [Editor Report · Acceptance letter]

8 Aug 2022

PONE-D-21-24368R2 

Factors associated with HIV viral suppression among adolescents in Kabale District, South Western Uganda. 

Dear Dr. Gordon:

I'm pleased to inform you that your manuscript has been deemed suitable for publication in PLOS ONE. Congratulations! Your manuscript is now with our production department. 

Kind regards, 

on behalf of

Dr. Siddappa N. Byrareddy 

Academic Editor

PLOS ONE